# Molecular typing and epidemiology profiles of human adenovirus infection among hospitalized patients with severe acute respiratory infection in Huzhou, China

**Deshun Xu, Liping Chen, Xiaofang Wu, Lei Ji** [ID] *

Huzhou Center for Disease Control and Prevention, Huzhou, China

* jileichn@163.com

## Abstract

### Background

Severe acute respiratory infections (SARI) threaten human health and cause a large number of hospitalizations every year. However, as one of the most common pathogen that cause acute respiratory tract infection, the molecular epidemiological information relating to human adenoviruses (HAdVs) among patients with SARI is limited. Here, we evaluate the epidemiological and molecular characteristics of HAdV infections among hospitalized patients with SARI from January 2017 to December 2019 in Huzhou, China.

### Methods

From January 2017 to December 2019, a total of 657 nasopharyngeal swabs collected from inpatients with SARI were screened for HAdV and other common respiratory viruses by multiplex real-time PCR. All samples that tested positive for HAdV were further typed by sequencing partial sequences of hexon gene. Genotypes of HAdV were confirmed by phylogenetic analysis. Epidemiological data were analyzed using Microsoft Excel 2010 and service solutions (SPSS) 21.0 software.

### Results

251 (38.20%) samples were positive for at least one respiratory virus. HAdV was the second common viral pathogen detected, with a detection rate of 7.08%. Infection with HAdV was found in all age groups tested (0<2, 2<5, 5<15, 15<50, 50<65, ≥65). Children under 15 years old accounted for 84.62% (44/52) of the infections. Higher activity of HAdV infection could be seen in spring-early autumn season. Seven different types of HAdV belonging to 4 species (HAdV-A, B, C, E) were identified in hospitalized SARI cases, with HAdV-B3 as the most prevalent HAdV types, followed by HAdV-B7 and HAdV-E4. HAdV-B3 was the most frequently detected genotype in 2017 and 2019, accounting for 75.00% (9/12) and 63.64% (7/11) of typed HAdV infections in 2017 and 2019, respectively. No predominant strain was

**Data Availability Statement:** The sequences of the HAdV strains obtained in this study were both deposited in the GenBank under the accession

numbers MW594169-MW594198 and the data is publicly available.

**Funding:** This work was supported by grants from Natural Science Foundation of Huzhou Science and Technology Bureau (grant number: 2021YZ23), the funders had no role in study design, data collection and analysis, decision to publish, or preparation of the manuscript.

**Competing interests:** The authors have declared that no competing interests exist.

responsible for HAdV infections in 2018, although HAdV-B7 (28.57%, 2/7) and HAdV-C1 (28.57%, 2/7) were the major causative genotypes.

## Conclusions

This study revealed the prevalence and the molecular epidemiological characteristics of HAdV infections among hospitalized patients with SARI in Huzhou from January 2017 to December 2019. The HAdV prevalence is related to age and season. As the most prevalent HAdV types, HAdV-B3 was co-circulating with other types and presented an alternate prevalence pattern.

## Background

Human adenoviruses (HAdVs) are non-enveloped, double-stranded DNA viruses belonging to the genus Mastadenovirus of the Adenoviridae family [1]. HAdVs have been recognized as pathogens that cause a broad spectrum of diseases, including respiratory illness, keratoconjunctivitis, gastroenteritis, cystitis and meningoencephalitis [2, 3]. They are associated with sporadic infection, as well as with community and institutional outbreaks. As a significant causative agent of respiratory tract illnesses, HAdV accounts for at least 5 to 10% of pediatric and 1 to 7% of adult respiratory tract infections (RTIs) [4, 5].

There are currently seven different HAdV species (HAdV-A through HAdV-G), and to date, 51 serotypes and over 70 genotypes have been identified based on serology, phylogenetic analyses and whole genomic sequencing (http://hadvwg.gmu.edu/). Different types of HAdVs display different tissue tropisms that correlate with clinical manifestations of infection. The HAdV types most commonly associated with respiratory infection belong to HAdV species B (HAdV-3, HAdV-7, HAdV-11, HAdV-14, HAdV-21), HAdV species C (HAdV-C1, -C2, -C5, and -C6) and HAdV species E (HAdV-4) [3].

The predominant types of HAdV circulating at a given time differ among countries or regions and change over time. Replacement of dominant viruses by new strains may occur because of transmission of novel strains between countries. During the last decade, outbreaks of respiratory tract infections caused by novel HAdV srains have occurred frequently in many countries including China [6, 7]. Therefore, clarifying the genotype of HAdV currently circulating is essential for epidemiological surveillance and a better understanding of the epidemic pattern of HAdV infection. At present, China has not yet established a national HAdV surveillance system. Although data about HAdV associated with respiratory infection in China can be found in several studies, most studies are performed with specific groups, especially for children [8–13]. There is a lack of epidemiological analyses of HAdV associated respiratory infection among patients in all age groups in China. The aim of this study was to evaluate the epidemiological and molecular characteristics of HAdV infections among hospitalized patients with severe acute respiratory infection (SARI) from January 2017 to December 2019 in Huzhou, a medium-sized city located in eastern China.

## Materials and methods

### Ethics statement

This study was part of the national SARI surveillance program and was approved by the human research ethics committee of Huzhou Center for Disease Control and Prevention. The

only human materials used were nasopharyngeal swabs collected from patients for routine detection. Data records and collected clinical specimens were deidentified and anonymous. Oral informed consents were obtained from each participant.

## Patients and specimens

During the influenza A H1N1 epidemic in 2009, a surveillance system for SARI was established to monitor influenza infection in China. As local SARI surveillance sentinel hospital, the First People's Hospital of Huzhou was responsible for sample collection from patients. The inclusion criteria for hospitalized SARI cases were as follows: the onset of the disease has a history of fever ($>$ 38˚C), accompanied by cough, and the onset does not exceed 10 days. Nasopharyngeal swabs were freshly collected and sent to Huzhou Center for Disease Control and Prevention for routine detection. All the specimens were stored at $-$ 80˚C until further processing. Demographic and clinical data were obtained from the hospital's database.

## Detection of HAdV and other common respiratory viruses

Total viral nucleic acids (DNA and RNA) were extracted from 200 μL of each specimen using TIANLONG Ex Viral DNA/RNA Kit (TIANLONG Biotech, Xi'an, China) according to the manufacturer's instructions. Multiplex real-time PCR kit (BioGerm, Shanghai, China) was used to detect HAdV and other common respiratory virus pathogens, including Human Influenza virus (HIFV), Human respiratory syncytial virus (HRSV), Human rhinovirus (HRV), Human bocavirus (HBOV), Human metapneumovirus (HMPV), Human Parainfluenza Virus (HPIV) type 1–4 and Human coronavirus (HCoV). The qPCR cycling program was as follows: 50˚C for 10 min, 95˚C for 5min, followed by 40 cycles of 95˚C for 10 s, and 55˚C for 40 s. Samples with a cycle threshold (Ct) $<$ 35 were regarded as positive.

## HAdV genotyping

HAdV-positive samples were further molecularly typed by nested PCR amplification and sequencing of HAdV hexon gene hyper-variable regions 1–6 (HVR1−6) as described previously [14]. Primer set AdhexF1 (nt 19135–19160; 5′-TICTTTGACATICGIGGGIGTICTIGA-3′) and AdhexR1 (nt 20009–20030; 5′-CTGTCIACIGCCTGRTTCCACA-3′) were used for first-round amplification; a second-round PCR was performed using primer set AdhexF2 (nt 19165–19187; 5′-GGYCCYAGYTTYAARCCCTAYTC-3′) and AdhexR2 (nt 19960–19985; 5′-GGTTCTGTCICCCAGAGARTCIAGCA-3′) if insufficient DNA was amplified from the first reaction for sequencing. The PCR products were visualized by electrophoresis and sent to TaKaRa Biotechnology (Dalian, China) for further purication and sequencing.

## Phylogenetic analysis

Partial nucleotide sequences of hexon gene obtained in this study were compared with the NCBI GenBank database (http://www.ncbi.nlm.nih.gov) by using online BLAST tools to preliminarily determine the genotype. Multiple sequence alignment and phylogenetic analysis were conducted using MEGA software version 6.06. The phylogenetic tree was generated using the neighbor-joining method and bootstrap analysis was performed with 1000 replications.

## Statistical analysis

Epidemiological data were analyzed using Microsoft Excel 2010 and service solutions (SPSS) 21.0 software. Statistical differences were determined using the Chi-square test and P-values $<$0.05 were considered to represent a statistically significant difference.

### Accession numbers

The partial hexon gene sequences obtained in this study have been deposited in GenBank under the accession numbers MW594169-MW594198.

## Results

### Characteristics of the SARI cases and the viral infection profiles

From January 2017 to December 2019, a total of 657 specimens (191 in 2017, 204 in 2018 and 262 in 2019) were collected from inpatients with SARI during the study period, of whom 361 (54.95%) were male and 296 (45.05%) were female. The age range was from 1 month to 86 years old with 590 (89.80%) cases being individuals younger than 15 years old.

The viral infection profiles are shown in Table 1. Overall, 251 (38.20%) samples were positive for at least one respiratory virus, the detection rate of respiratory virus was 45.54% (87/191) in 2017, 36.27% (74/204) in 2018 and 34.35% (90/262) in 2019. During the study period, the most commonly detected viral pathogen in SARI cases was RSV, with a prevalence rate of 10.65% (70/657), followed by HAdV (7.91%, 52/657) and HIFV (6.09%, 40/657). HMPV was detected in 30 patients (4.57%), HPIV was detected in 24 (3.65%), HBOV was detected in 21 (3.20%), HRV was detected in 11 (1.67%), and HCoV was detected in 3 patients (0.46%).

### Epidemiology of HAdV

Among the 52 HAdV-infected patients, 31 (58.33%) were male and 21 (41.67%) were female (Table 2). No significant difference was observed in males and females in the HAdV-infected cases (P = 0.481). Infection with HAdV was found in all age groups tested (0<2, 2<5, 5<15, 15<50, 50<65, ≥65). Children under 15 years old accounted for 84.62% (44/52) of the

**Table 1. Viral infection profiles in hospitalized patients with SARI in Huzhou, 2017–2019.**

| Years | SARI cases | Any viral etiology | Viral infection profiles N (%) | | | | | | | |
|-------|-----------|--------------------|------------|-----------|-----------|-----------|-----------|-----------|-----------|-----------|
| | | | HRSV | HAdV | HIFV | HMPV | HPIV | HBOV | HRV | HCoV |
| 2017 | 191 | 87 | 32 (16.75) | 18 (9.42) | 8 (4.19) | 8 (4.19) | 10 (5.24) | 8 (4.19) | 3 (1.57) | 0 (0.00) |
| 2018 | 204 | 74 | 17 (8.33) | 8 (3.92) | 21 (10.29) | 14 (6.86) | 5 (2.45) | 7 (3.43) | 2 (0.98) | 0 (0.00) |
| 2019 | 262 | 90 | 21 (8.02) | 26 (9.92) | 11 (4.20) | 8 (3.05) | 9 (3.44) | 6 (2.29) | 6 (2.29) | 3 (1.15) |
| Total | 657 | 251 | 70 (10.65) | 52 (7.91) | 40 (6.09) | 30 (4.57) | 24 (3.65) | 21 (3.20) | 11 (1.67) | 3 (0.46) |

**Table 2. HAdV-positive in hospitalized patients of different ages and gender with SARI.**

| Variable | Tested SARI cases N (percentage) | HAdV-positive cases N (percentage) | HAdV-negtive cases N (percentage) | Positive rate | $\chi^2$ | P |
|----------|----------------------------------|-------------------------------------|------------------------------------|---------------|----------|----|
| Sex | | | | | 0.497 | 0.481 |
| Male | 361(53.40) | 31(58.33) | 330(54.55) | 8.59% | | |
| Female | 296(46.60) | 21(41.67) | 275(45.45) | 7.09% | | |
| Age (years) | | | | | 0.431 | 0.476 |
| 0<2 | 99(15.07) | 5(9.62) | 94(16.55) | 5.05% | | |
| 2<5 | 180(27.40) | 17(32.69) | 163(23.51) | 9.44% | | |
| 5<15 | 241(36.68) | 22(42.31) | 219(43.05) | 9.13% | | |
| 15<50 | 70(10.65) | 5(9.61) | 65(6.95) | 7.14% | | |
| 50<65 | 32(4.87) | 1(1.92) | 31(4.64) | 3.13% | | |
| ≥65 | 35(5.33) | 1(1.92) | 34(5.30) | 2.86% | | |
| Total | 657 | 52 | 605 | 7.91% | | |

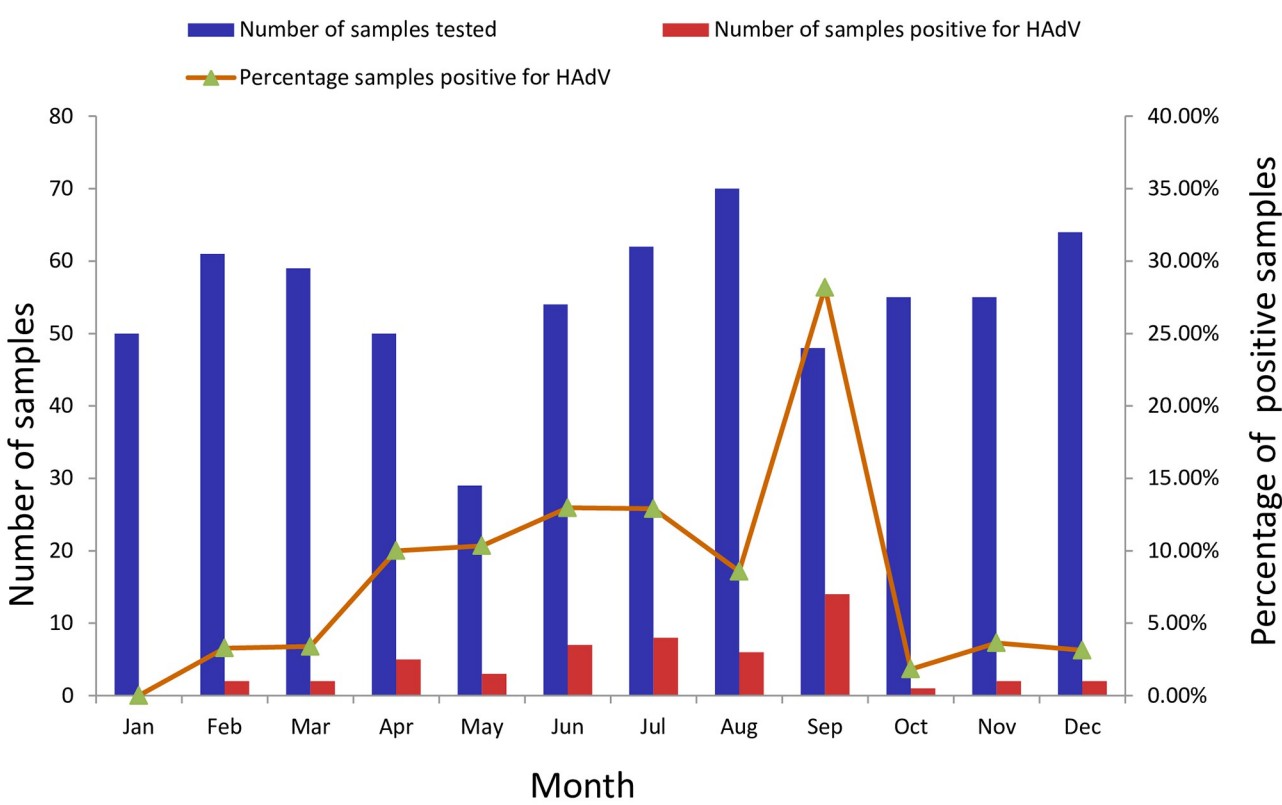

**Fig 1. Monthly distribution of HAdV infections from January 2017 to December 2019.**

infections. There were no significant differences in HAdV detection rates among different age groups (P = 0.467). The highest detection rate was in the 2<5 year age group (9.44%), followed by 5<15 years (9.13%), 15<50 years (7.14%), 0<2 years (5.05%), 50<65 years (3.13%) and ≥65 years (2.86%).

HAdV detection rate varied from year to year, from 9.42% (18/191) in 2017, 3.92% (8/204) in 2018 to 9.92% (26/262) in 2019 (Table 1). The monthly distribution of HAdV infections is shown in Fig 1. HAdV was detected in every month throughout the study period except January. Higher activity of HAdV infection could be seen from spring to early autumn (April to September), and the detection rate in September reached a peak of 28.17%. In contrast, lower activity of HAdV infection were observed during late autumn to winter (from October to February), when the average detection rate was only 2.37%.

Additionally, 13.46% (n = 7) of the 52 HAdV-infected cases were co-detected with other respiratory pathogens. RSV (n = 3) was the most frequently co-detected virus. HPIV (n = 2), HMPV (n = 1) and HRV (n = 1) were also found to be co-infected with HAdV.

## HAdV genotyping and phylogenetic analysis

Of the 52 HAdV-positive samples confirmed by real-time RT-PCR, 30 samples were successfully sequenced and genotyped by nested-PCR. Phylogenetic analysis based on partial hexon sequences indicated that 4 species (A, B, C, E) of HAdV, including 7 different types were identified throughout the study period (Fig 2). HAdV-B3 (n = 17, 56.67%) was the most prevalent HAdV types, followed by HAdV-B7 (n = 5, 16.67%) and HAdV-E4 (n = 3, 10.00%). HAdV-C1 (n = 2, 6.67%), HAdV-C2 (n = 1, 3.33%), HAdV-B21 (n = 1, 3.33%) and HAdV-B55 (n = 1,

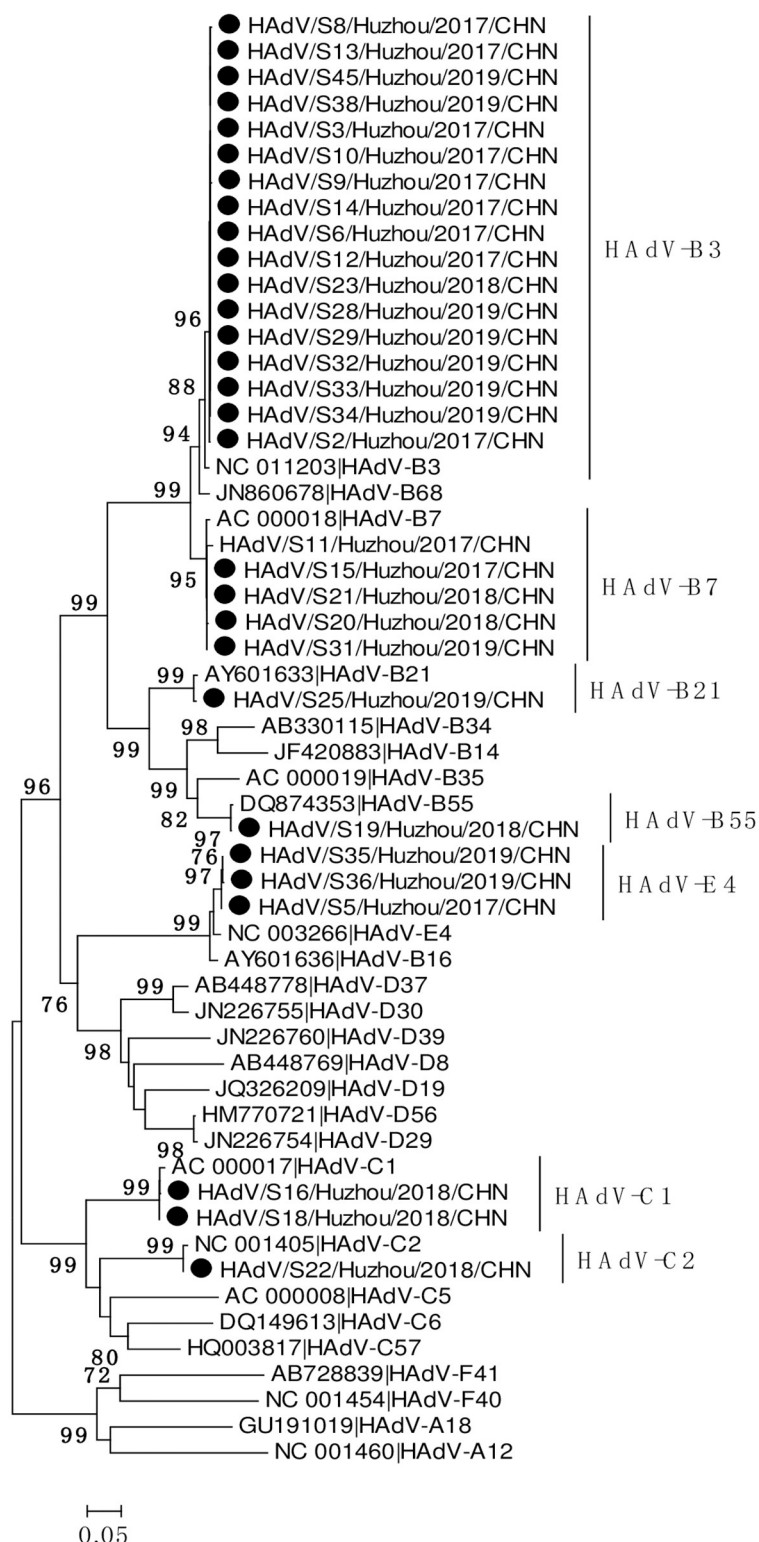

**Fig 2. Phylogenetic analyses based on partial hexon sequences of HAdV strains.** The trees were generated using the neighbor-joining method, validated by 1000 bootstrap replicates. Bootstrap values ≥ 70% are shown on the branch. HAdV sequences identified in this study are indicated by closed circles.

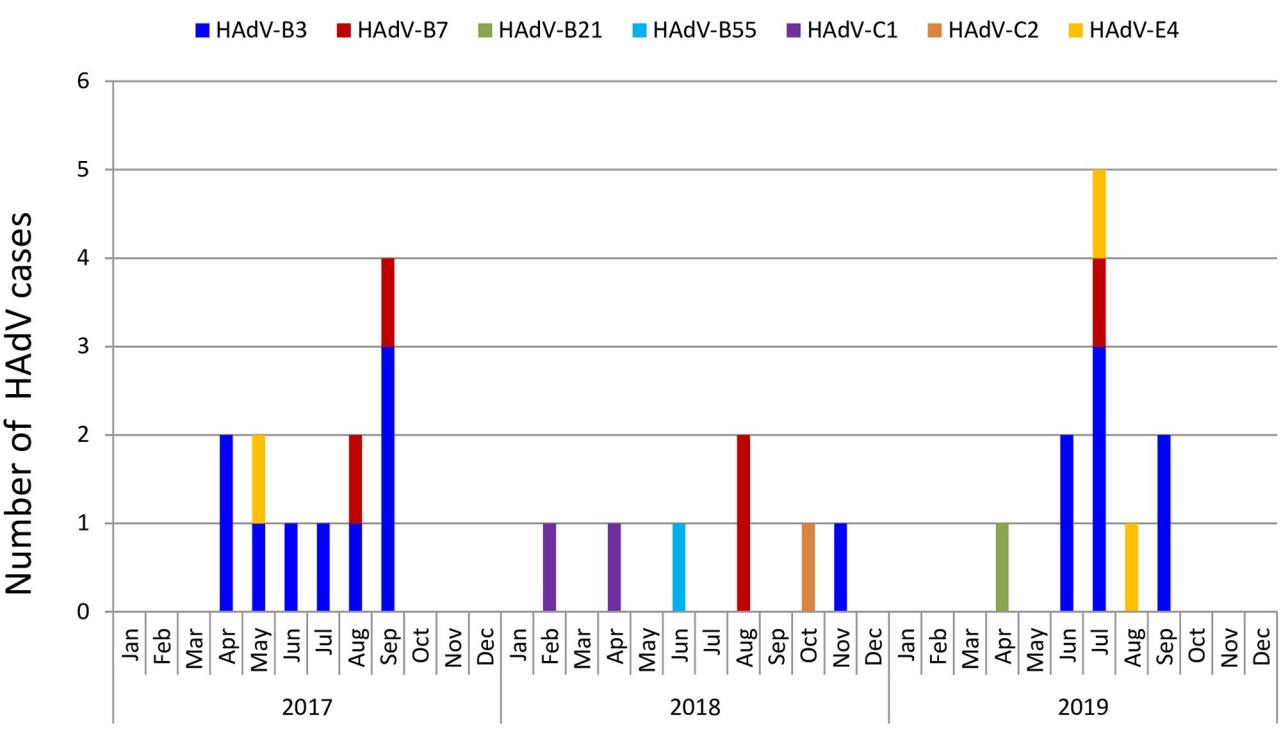

**Fig 3. Distribution of HAdV genotypes detected according to month.**

3.33%) were also detected. The genotype distribution of HAdV infections in each month is shown in Fig 3. The predominant genotypes of HAdV during our study period varied according to surveillance year. Overall, HAdV-B3 was the most frequently detected genotype in 2017 and 2019, accounting for 75.00% (9/12) and 63.64 (7/11) of typed HAdV infections, respectively. Five different types were detected in 2018, including HAdV-B7 (n = 2), HAdV-C1 (n = 2), HAdV-B3 (n = 1), HAdV-B55 (n = 1) and HAdV-C2 (n = 1). No predominant strain was responsible for HAdV infections in 2018, although HAdV-B7 (28.57%, 2/7) and HAdV-C1 (28.57%, 2/7) were the major causative genotypes.

## Discussion

SARI is one of the most common diseases in human and the leading cause of hospitalization in children worldwide [15, 16]. Because the early clinical symptoms of respiratory infections caused by viruses are similar, and the imaging findings lack specificity, pathogen detection is very important in clinical diagnosis and epidemiological monitoring. The present study was carried out from January 2017 to December 2019 among hospitalized patients with SARI in Huzhou, China. During the study period, a total of 657 hospitalized SARI cases were enrolled, of which 80.57% were children under 15 years of age. These results suggest that SARI is still an important factor affecting the health of local children. In total, 38.20% of hospitalized SARI cases in our study exhibited at least one respiratory virus, which was consistent with previous reports from China (33.44%-41.50%) [17, 18] and other countries (37.57%-41.8%) [19–21].

HAdV was the second most common viral pathogen detected, with a detection rate of 7.08%, which is lower than the finding in SARI cases of hospitalized children in Beijing (11.90%) and Shanghai (14.70%) [22]. Previous studies have indicated that HAdV is the major pathogen that causes respiratory tract infections in children, especially for children younger

than 5 years [8, 10]. As expected, we found that HAdV infection mainly occurred in children under 15 years of age (84.62%), and the detection rate reached a peak (9.44%) in children aged 2 to <5 years.

Most respiratory viral infections have seasonality, of note, this seasonality might vary according to geographical location. Price RHM *et al.* have investigated the relationship between meteorological factors and viral seasonality in Scotland over a 6.5-year period [23]. In their study, HAdV is present throughout the year without a clear seasonality and prefer temperatures around 9˚C. With regards to HAdV in southern Brazil, it is more frequent during the winter, however, in relation to other respiratory viruses, HAdV is more prevalent in summer [21]. Previous studies have shown that the epidemic peak seasons of HAdV-associated respiratory infections varies in different parts of China, and even in different monitoring years in the same region. Our study revealed that HAdV showed higher activity in the relatively high temperature seasons (spring to early autumn), which is similar to what has been found in Beijing (Northern China) [8] and Guangzhou (Southern China) [13], where HAdV infections occurred throughout the year with the highest prevalence in the summer. However, this finding is discordant with other studies conducted in Northern China that have reported seasonal peaks for HAdV infections in winter and spring [9, 12]. It is worth mentioning that the surveillance period of the above-mentioned studies conducted in different regions of China varies, and the predominant HAdV types circulated are also different. A recent study from Hunan indicates that different HAdV types showed a different seasonal distribution patterns: HAdV-3 was the predominant type of HAdV infection during summer, while HAdV-7 had the highest detection rate during spring [11]. Based on the above research, we speculate that the discrepant seasonal peak for HAdV infections are not only related to regional differences, but also related to the major types of HAdV circulating locally.

Globally the HAdV types most commonly associated with respiratory syndromes belong to HAdV species B, C or E. Many studies have reported that HAdV–B3, HAdV–B7 and HAdV–C2 are the most prevalent types in China, but the predominant type distribution vary among different regions and change over time. For example, most of HAdV-positive cases were caused by HAdV-B3 from 2012 to 2013 in Southern China [13], while HAdV-B7 dominated in Northern China during the same study period [10]. However, recent reports indicated that the most predominant types have changed into HAdV-B3 and HAdV-C2 in some Northern cities of China in 2017–2018 [8, 9]. Throughout the present study period, seven different types of HAdV belonging to four species (HAdV-A, B, C, E) were identified in hospitalized SARI cases, with HAdV-B3 as the most prevalent HAdV types, followed by HAdV-B7 and HAdV-E4. Our monitoring data showed that no type of HAdV presented absolutely predominant during HAdV epidemic seasons, HAdV-B3 was co-circulating with other types and presented an alternate prevalence pattern. Overall, HAdV-B3 was the most frequently detected genotype in 2017. No predominant strain was responsible for HAdV infections in 2018, with HAdV-B7 and HAdV-C1 as the major causative genotypes. HAdV-B3 re-emerged as the predominant genotype in 2019. Similar epidemic pattern were observed in a prolonged surveillance study conducted in southeastern China, where HAdV-7 and HAdV-3 alternate as the predominant genotypes causing pediatric pneumonia [24]. It is worth noting that in 2017 and 2019, when HAdV-3 presented as the predominant type detected, the detection rate of HAdV was significantly higher than that in 2018 (9.42% in 2017, 3.92% in 2018 and 9.92% in 2019). The reasons need be further explored. During HAdV infection, neutralizing antibodies are formed against the epitopes located in the hyper variable regions (HVRs) of the hexon protein. Just recently, Haque E et al. explored the variation in HVRs of hexon among globally distributed strains of HAdV-3 [25]. They found that the HVRs of HAdV-3 strains circulating worldwide were highly heterogeneous and have been mutating continuously since their original

isolation and suggested that, this heterogeneity may explain the worldwide increased prevalence of HAdV-3 respiratory infections.

Recent HAdV epidemiology studies showed that there was very high co-infection rate between HAdV and other pathogens in respiratory tract infection cases (37.50%-74.85%) [8, 9, 11]. In our study, coinfection of HAdVs and other respiratory viruses was only detected in 13.46% of the SARI cases. Such discrepant co-infection rate may be caused by the different selection criteria of the research objects and methodological differences.

Our study is limited by a single-site setting, small sample size, and especially the partial genotyping of detected HAdVs. Genotyping was only successful for 57.69% (30/52) of HAdV infection cases. Besides, typing of HAdV was merely performed by sequencing of partial hexon gene in the present study, which is hard to find any potential recombination between different types of HAdV strains.

## Conclusions

In conclusion, this study revealed the prevalence and molecular epidemiological characteristics of HAdV infections among hospitalized patients with SARI in Huzhou from January 2017 to December 2019. HAdV was the second common viral pathogen detected in SARI cases, with most (84.62%) HAdV-positives cases detected among children < 15 years of age. Higher activity of HAdV infection could be seen in spring -early autumn season. As the most prevalent HAdV types, HAdV-B3 was co-circulating with other types and presented an alternate prevalence pattern. Our results provide a reliable scientific basis to better understand the role played by HAdVs in SARI cases, and for the prevention and control of HAdV infection.

## Supporting information

**S1 File.**
(FAS)

## Acknowledgments

We thank the staff of the First People's Hospital in Huzhou for collecting the samples.

## Author Contributions

**Investigation:** Deshun Xu.

**Methodology:** Deshun Xu, Liping Chen, Xiaofang Wu, Lei Ji.

**Project administration:** Lei Ji.

**Writing – original draft:** Lei Ji.

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
