## [Decision Letter · Decision Letter 0]

6 Oct 2021

PONE-D-21-05692

Molecular Typing and Epidemiology Profiles of Human Adenovirus Infection among Hospitalized Patients with Severe Acute Respiratory Infection in Huzhou, China

PLOS ONE

Dear Dr. Lei Ji,

Thank you for submitting your manuscript to PLOS ONE. After careful consideration, we feel that it has merit but does not fully meet PLOS ONE’s publication criteria as it currently stands. Therefore, we invite you to submit a revised version of the manuscript that addresses the points raised during the review process.

Please read the reviewers´comments.

Please submit your revised manuscript by  September 10th. If you will need more time than this to complete your revisions, please reply to this message or contact the journal office at plosone@plos.org. Please include the following items when submitting your revised manuscript:

We look forward to receiving your revised manuscript.

Kind regards,

Rosa Maria Wong-Chew, MD, DSc

Academic Editor

PLOS ONE

Journal Requirements:

2. Please complete all items on the Clinical Studies Checklist that are relevant for your submission, by following this link: http://journals.plos.org/plosone/s/file?id=dc11/PLOSOne_Clinical_Studies_Checklist.docx (Contact us at plosone@plos.org if you cannot access the document.) There may be overlap between the checklist items and other queries listed below; please address any duplicated queries both in your response email and on the checklist itself. Upload the completed Clinical Studies Checklist as file type “Other” when you re-submit your manuscript. This document is for internal journal use only and will not be published if your article is accepted. The requested information will help us to assess whether your submission complies with PLOS ONE’s policies and adheres to applicable reporting standards. Note that your manuscript may be rejected if you provide incomplete or inadequate responses to the checklist questions and that changing the ‘Section/Category’ of your article does not affect this requirement.

Reviewers' comments:

Reviewer's Responses to Questions

**Comments to the Author**

1. Is the manuscript technically sound, and do the data support the conclusions?

Reviewer #1: Yes

Reviewer #2: Yes

2. Has the statistical analysis been performed appropriately and rigorously? 

Reviewer #1: Yes

Reviewer #2: Yes

3. Have the authors made all data underlying the findings in their manuscript fully available?

Reviewer #1: Yes

Reviewer #2: Yes

4. Is the manuscript presented in an intelligible fashion and written in standard English?

Reviewer #1: Yes

Reviewer #2: Yes

5. Review Comments to the Author

Reviewer #1: The study assessed cases of severe acute respiratory infection in a hospital in Huzhou, China, for epidemiological analysis of human adenovirus (HAdV), including viral genotyping based on sequencing of the hexon gene and nested-PCR. Results add more information about the epidemiology of HAdV in China, complementing previous studies in the country. Considering that HAdV is an important respiratory virus worldwide, I suggest discussing the findings of the study with studies performed in other countries, such as in Europe (Price RHM et al., https://doi.org/10.1038/s41598-018-37481-y) and South America (Pscheidt et al., DOI: 10.1002/rmv.2189).

There are several English and typing details that should be revised and corrected, as pointed in the pdf file. For example, symbols such as ~ to describe age group (0～, 2～, 5～,15～, 50～, 65～) does not make much sense. Please use 0≤2; 2≤5; 5≤15, etc.

In the methodology, I believe there is a step missing in the PCR description. Isn't there an extension step at 72ºC after the annealing step?

In Table 1, I suggest including the percentage in addition to the number of cases – N (%).

In Table 2, use Sex instead of Gender.

In the Results section, it is necessary to make a thorough revision of typing. Spaces missing, double spaces, typos etc. make it difficult to read the results, mainly between lines 192–209.

When discussing HAdV seasonality, I suggest comparing the study with those from other geographic regions, as viral circulation has been associated with temperature and circulation of other respiratory viruses.

Other comments and suggestions can be found in the pdf file (attached).

Reviewer #2: Comments to the Author

Xu et al., report valuable information on HAdV epidemiology including prevalence, seasonality, and molecular epidemiology in patients with SARI in China. The study methods are sound and the results are an important contribution to the field. I have only one comment: The authors were unable to type up to 42.3% (22/52) of the study samples. What could explain this high failure rate.

6. PLOS authors have the option to publish the peer review history of their article (what does this mean?). If published, this will include your full peer review and any attached files.

Reviewer #1: **Yes: **Ana Beatriz Gorini da Veiga

Reviewer #2: No

---

## [Author Response · Author response to Decision Letter 0]

3 Nov 2021

Responses to the reviewers’ comments: 

(Q as comments, A as our responses)

Reviewer 1

Q: Considering that HAdV is an important respiratory virus worldwide, I suggest discussing the findings of the study with studies performed in other countries, such as in Europe (Price RHM et al., https://doi.org/10.1038/s41598-018-37481-y) and South America (Pscheidt et al., DOI: 10.1002/rmv.2189). When discussing HAdV seasonality, I suggest comparing the study with those from other geographic regions, as viral circulation has been associated with temperature and circulation of other respiratory viruses.

A: Thanks for the reviewer’s suggestion. The two references mentioned above have been added (see line 230, 239,242). “Most respiratory viral infections have seasonality, of note, this seasonality might vary according to geographical location. Price RHM et al. have investigated the relationship between meteorological factors and viral seasonality in Scotland over a 6.5-year period. In their study, HAdV is present throughout the year without a clear seasonality and prefer temperatures around 9 °C. In another study conducted in patients with respiratory infection in southern Brazil, HAdV circulated year‐round, with higher frequency during winter and early spring.” We have added these sentences to the manuscript, see line 238-245.

Q: There are several English and typing details that should be revised and corrected, as pointed in the pdf file. For example, symbols such as ~ to describe age group (0～, 2～, 5～,15～, 50～, 65～) does not make much sense. Please use 0≤2; 2≤5; 5≤15, etc. 

A: Thanks for the reviewer’s suggestion. Corrections have been made in the revised version.

Q: In the methodology, I believe there is a step missing in the PCR description. Isn't there an extension step at 72ºC after the annealing step?

 A: According to the multiplex real-time PCR kit’s instructions, the qPCR cycling program was as 50 °C for 10 min, 95 °C for 5min, followed by 40 cycles of 95 °C for 10 s, and 55 °C for 40 s. The same temperature (55 °C) is used for the annealing and extension steps.

Q: In Table 1, I suggest including the percentage in addition to the number of cases – N (%).

In Table 2, use Sex instead of Gender.

A: Thanks for the reviewer’s suggestion. Corrections have been made in the revised version .

Q: Other comments and suggestions can be found in the pdf file (attached).

A: Thanks. Corrections have been made in the revised version according to comments and suggestions in the pdf file.

Reviewer 2

Q: I have only one comment: The authors were unable to type up to 42.3% (22/52) of the study samples. What could explain this high failure rate.

A: Thanks for the reviewer’s suggestion. Since the sensitivity of ordinary RT-PCR is lower than that of real-time RT-PCR, more accurate genotyping was only possible for 57.7% (30/52) of the HAdV-positive samples confirmed by real-time RT-PCR, the remaining 22 samples with most of them got a cycle threshold (Ct) ≥ 30 were failed to genotyped.

---

## [Decision Letter · Decision Letter 1]

3 Feb 2022

PONE-D-21-05692R1Molecular Typing and Epidemiology Profiles of Human Adenovirus Infection among Hospitalized Patients with Severe Acute Respiratory Infection in Huzhou, ChinaPLOS ONE

Dear Dr. Ji,

Thank you for submitting your manuscript to PLOS ONE. After careful consideration, we feel that it has merit but does not fully meet PLOS ONE’s publication criteria as it currently stands. Therefore, we invite you to submit a revised version of the manuscript that addresses the points raised during the review process.

Specifically, please address Reviewer 1's remaining concerns.

We look forward to receiving your revised manuscript.

Kind regards,

Jianhong Zhou

Associate Editor

PLOS ONE

Journal Requirements:

Reviewers' comments:

Reviewer's Responses to Questions

**Comments to the Author**

1. If the authors have adequately addressed your comments raised in a previous round of review and you feel that this manuscript is now acceptable for publication, you may indicate that here to bypass the “Comments to the Author” section, enter your conflict of interest statement in the “Confidential to Editor” section, and submit your "Accept" recommendation.

Reviewer #1: (No Response)

Reviewer #2: All comments have been addressed

2. Is the manuscript technically sound, and do the data support the conclusions?

Reviewer #1: Yes

Reviewer #2: Yes

3. Has the statistical analysis been performed appropriately and rigorously? 

Reviewer #1: Yes

Reviewer #2: Yes

4. Have the authors made all data underlying the findings in their manuscript fully available?

Reviewer #1: Yes

Reviewer #2: Yes

5. Is the manuscript presented in an intelligible fashion and written in standard English?

Reviewer #1: Yes

Reviewer #2: Yes

6. Review Comments to the Author

Reviewer #1: The authors made significant changes in the manuscript according to the reviewers' comments. There are still a few things to correct regarding typing and other details.

For example, for age groups 0≤2, 2≤5, 5≤15 etc. means that 2 years old are included in both groups 0-2 and 2-5. The correct is 0<2, 2<5, 5<15 and so on.

With regards to HAdV in southern Brazil, it is more frequent during the winter, however, in relation to other respiratory viruses, HAdV is more prevalent in summer (Pscheidt et al, 2021).

Other comments were made in the pdf file, attached.

Reviewer #2: Nothing to report, all my comments have been adressed favorably.

Thanks for the invitation.

Best regards.

7. PLOS authors have the option to publish the peer review history of their article (what does this mean?). If published, this will include your full peer review and any attached files.

Reviewer #1: No

Reviewer #2: No

---

## [Author Response · Author response to Decision Letter 1]

2 Mar 2022

Dear Editor Jianhong Zhou,

Thank you very much for giving us an opportunity to revise our manuscript. We also appreciate reviewers very much for their positive and constructive comments and suggestions on our manuscript entitled “Molecular Typing and Epidemiology Profiles of Human Adenovirus Infection among Hospitalized Patients with Severe Acute Respiratory Infection in Huzhou, China” (PONE-D-21-05692).

We have carefully addressed all of the comments from the reviewers, as outlined in the point-by-point responses attached below. We hope that you find the revised manuscript now acceptable for publication in PLOS ONE.

Updated statement: This work was supported by grants from Natural Science Foundation of Huzhou Science and Technology Bureau (grant number: 2021YZ23), the funders had no role in study design, data collection and analysis, decision to publish, or preparation of the manuscript.

Yours Sincerely 

Lei Ji

Responses to the reviewers’ comments: 

(Q as comments, A as our responses)

Reviewer 1

Q: for age groups 0≤2, 2≤5, 5≤15 etc. means that 2 years old are included in both groups 0-2 and 2-5. The correct is 0<2, 2<5, 5<15 and so on.

A: Thanks for the reviewer’s suggestion. Corrections have been made in the revised version.

Q: With regards to HAdV in southern Brazil, it is more frequent during the winter, however, in relation to other respiratory viruses, HAdV is more prevalent in summer 

A: Thanks for the reviewer’s suggestion. Corrections have been made in the revised version, see line 237-241.

Q: Other comments and suggestions can be found in the pdf file (attached).

A: Thanks. Corrections have been made in the revised version according to comments and suggestions in the pdf file.

---

## [Decision Letter · Decision Letter 2]

14 Mar 2022

Molecular Typing and Epidemiology Profiles of Human Adenovirus Infection among Hospitalized Patients with Severe Acute Respiratory Infection in Huzhou, China

PONE-D-21-05692R2

Dear Dr. Ji,

We’re pleased to inform you that your manuscript has been judged scientifically suitable for publication and will be formally accepted for publication once it meets all outstanding technical requirements, including a couple of minor corrections indicated by reviewer no. 1.

Kind regards,

Luis Menéndez-Arias, Ph. D.

Academic Editor

PLOS ONE

Additional Editor Comments (optional):

Reviewers' comments:

Reviewer's Responses to Questions

**Comments to the Author**

1. If the authors have adequately addressed your comments raised in a previous round of review and you feel that this manuscript is now acceptable for publication, you may indicate that here to bypass the “Comments to the Author” section, enter your conflict of interest statement in the “Confidential to Editor” section, and submit your "Accept" recommendation.

Reviewer #1: All comments have been addressed

Reviewer #2: All comments have been addressed

2. Is the manuscript technically sound, and do the data support the conclusions?

Reviewer #1: Yes

Reviewer #2: Yes

3. Has the statistical analysis been performed appropriately and rigorously? 

Reviewer #1: Yes

Reviewer #2: Yes

4. Have the authors made all data underlying the findings in their manuscript fully available?

Reviewer #1: Yes

Reviewer #2: Yes

5. Is the manuscript presented in an intelligible fashion and written in standard English?

Reviewer #1: Yes

Reviewer #2: Yes

6. Review Comments to the Author

Reviewer #1: There are only 2 corrections that have not beem made:

Line 64: it should be "prevalent HAdV type" (not types)

Line 232: should be "children aged 2 to 5 years", delete <

Reviewer #2: The study methods are sound and the results are an important contribution to the field. All my comments have been addressed by the authors.

7. PLOS authors have the option to publish the peer review history of their article (what does this mean?). If published, this will include your full peer review and any attached files.

Reviewer #1: No

Reviewer #2: No

---

## [Editor Report · Acceptance letter]

13 Apr 2022

PONE-D-21-05692R2 

Molecular Typing and Epidemiology Profiles of Human Adenovirus Infection among Hospitalized Patients with Severe Acute Respiratory Infection in Huzhou, China 

Dear Dr. Ji:

I'm pleased to inform you that your manuscript has been deemed suitable for publication in PLOS ONE. Congratulations! Your manuscript is now with our production department. 

Kind regards, 

on behalf of

Dr. Luis Menéndez-Arias 

Academic Editor

PLOS ONE